# Allostatic Control of Persistent States in Spiking Neural Networks for perception and computation

## Abstract

We introduce a novel model for updating perceptual beliefs about the environment by extending the concept of Allostasis to the control of internal representations. Allostasis is a fundamental regulatory mechanism observed in animal physiology that orchestrates responses to maintain a dynamic equilibrium in bodily needs and internal states. In this paper, we focus on an application in numerical cognition, where a bump of activity in an attractor network is used as a spatial-numerical representation. While existing neural networks can maintain persistent states, to date, there is no unified framework for dynamically controlling spatial changes in neuronal activity in response to environmental changes. To address this, we couple a well-known allostatic microcircuit, the Hammel model, with a ring attractor, resulting in a Spiking Neural Network architecture that can modulate the location of the bump as a function of some reference input. This localized activity in turn is used as a perceptual belief in a simulated subitization task – a quick enumeration process without counting. We provide a general procedure to fine-tune the model and demonstrate the successful control of the bump location. We also study the response time in the model with respect to changes in parameters and compare it with biological data. Finally, we analyze the dynamics of the network to understand the selectivity and specificity of different neurons to distinct categories present in the input. The results of this paper, particularly the mechanism for moving persistent states, are not limited to numerical cognition but can be applied to a wide range of tasks involving similar representations

## 1 Introduction

Neural networks in the brain are structured networks with a space-like structure, topologically analogous to a metric space (i.e. Euclidean) (Gerstner et al. (2014)). The use of these organized structure are found not only in concrete representations, such as head direction (Pisokas et al. (2020)), orientation and compass for foraging (Stentiford et al. (2024)), and place and grid cells (Whelan et al. (2022)), but also in more abstract topographic maps that represent emotions, relationships, and, as we show in this paper, numbers (Leslie et al. (2008)). Moreover, these structures extend beyond biological systems; they are increasingly being employed in artificial embodied agents, demonstrated through tasks such as numerical cognition through embodied learning mechanisms and on development of grounding transfer from concrete concept of sensorimotor experiences to abstract conceptual knowledge (Alessandro & Angelo (2021)).

A key property of these networks is their capacity to hold persistent states, or bumps of activity (Gerstner et al. (2014)), which are associated with the representation of information in all of the mentioned systems of concrete and abstract representations (Hopfield (2015)). In this paper, we explore one such key network, the ring attractor model, which uses a bump of activity that is localized to dynamically control spatial changes in neuronal activity (i.e., activities influenced by changes in surrounding contextual features) (Pisokas et al. (2020)). Although these networks can represent to track changes in the environment, this raises the question of how we can exert arbitrary control over the position of bump activity in a structured spiking neural network. We infer that a structured network (i.e., ring attractor models, place cell and grid cells, Kohonen maps) would more closely resemble the brain's functionality than continuous neural field models, for which results in these

continuous neural field models already exists (Gerstner et al. (2014)). Moreover, we pose a question on whether these bumps can represent abstract information that does not have explicit dynamics.

To approach the first question, we propose the use of self-regulation as a mechanism for controlling the bump activity towards a desired state. The self-regulation mechanism here considers a neuronal model of set-point thermo-regulation, i.e., Hammel Model (Boulant (2006)), where this acts as a negative feedback control system for the bump activity. Considering that self-regulation can achieve arbitrary control of manifold physiological variables, these variables have similar properties to the bump activity (i.e., there is always a "place" where the bumps are supposed to be). An important aspect of self-regulation is allostasis (Sterling (2012)), which extends the simpler concept of homeostasis (i.e. the maintenance of physiological set-points) to cover adaptive, predictive and competitive aspects of self-regulation. Thus, we further extend the usage of allostasis to tackle perception and computation in structured neural networks by demonstrating the predictive behaviors towards its desired state based on numerical cognition tasks.

Thus, we introduce subitization to approach the second question as a predictive task over numerical capability. We use subitizing as a model system to explore abstract properties with neither intrinsic nor explicit dynamics, yet with a putative metric representation: the number line (Leslie et al. (2008)). Subitization is a special mechanism capable of recognizing quantity strictly through perception without the need for counting (Dehaene & Cohen (1994)) and it can be characterized as a form of categorization (Benoit & Henri Lehalle (2004)). This requires the distinction of elements and their mapping to specific internal representations; in this case, bump activity is mapped to numerical representations. Since subitization tasks are associated with the prediction of discrete number representations, this form of representations can be modeled as an accumulator model with bi-directional mappings (Gallistel & Gelman (2000)). Thus, we can consider numerosity tasks, such as subitization, to exemplify the spatial-temporal control of these mappings. While there are other models that encode numerosity tasks into a single neuron network (Rapp & Martin Paul Nawrot (2020)), we use our allostatic network approach to achieve results that can match behavioral and neuronal findings.

## 1.1 MODEL DERIVATION

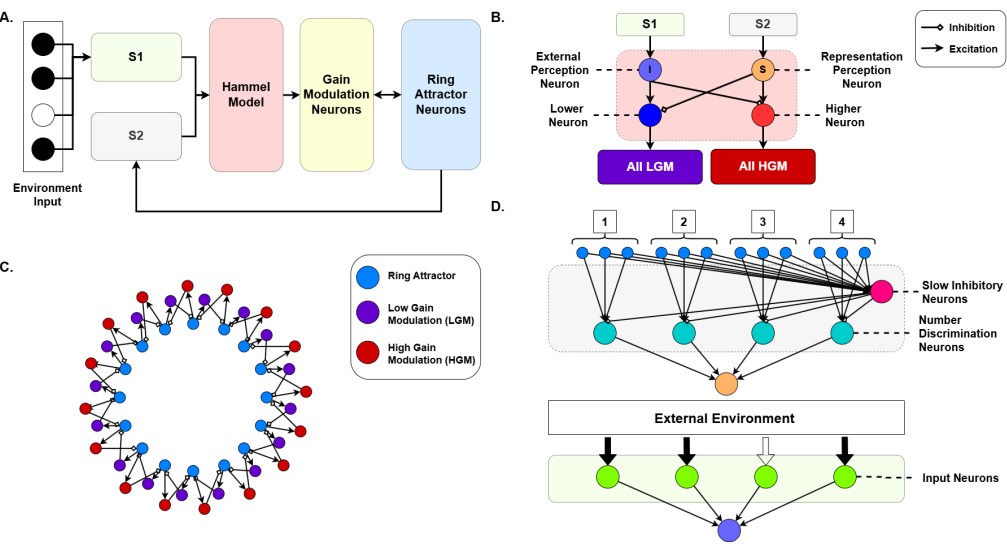

Figure 1: AlloNet model definition. **A.** Overview of model architecture. Two inputs are associated to homeostasis model: Environmental input (S1) and Ring attractor feedback (S2). **B.** Representation of how Hammel model is used in the architecture. Higher and lower neurons are associated to one-to-all connection onto high gain modulation (HGM) and low gain modulation (LGM) respectively. **C.** Synaptic connection management between the ring attractor and gain modulation neurons (LGM and HGM). **D.** Representation of S1 and S2. These define how information, both external input (bottom figure) and feedback input (top figure), is being represented into the network.

Table 1: Default neuron parameters from NEST simulator are used for the network. The following parameters are initially based on (Laing & Chow (2001)), where a parameter search is done to find the regime for the ring attractor drift to be linear to the Hammel input. In the case of changes made, these parameters are mentioned below and abbreviations used for parts of the model can be referred from Figure 1.

| Name | Description | Value |
|---|---|---|
| $V_{th}$ | Threshold Function | 0.8mV |
| $\tau^1_{syn\_ex}$ | Excitatory Synaptic Alpha Function(Hammel Model, S) | 75ms |
| $\tau^2_{syn\_ex}$ | Excitatory Synaptic Alpha Function (Ring Attractor) | 27ms |
| $\tau^3_{syn\_ex}$ | Excitatory Synaptic Alpha Function (HGM, LGM) | 1000ms |
| $\tau^4_{syn\_ex}$ | Excitatory Synaptic Alpha Function (Slow Inhibitory Neuron, Number Discrimination Neuron) | 100ms |
| $\tau^5_{syn\_ex}$ | Excitatory Synaptic Alpha Function (I) | 1ms |
| $\tau_{syn\_in}$ | Inhibitory Synaptic Alpha Function | 50ms |
| $I_e$ | Input Current (Default) | $0.45 \times 10^6$pA |
| $I^1_e$ | Input Current (HGM, LGM, Slow Inhibitory Neuron, Number Discrimination Neuron) | 0pA |
| $\tau_m$ | Membrane Time Constant (Default) | 7.04ms |
| $\tau^1_m$ | Membrane Time Constant (HGM, LGM) | 4.2ms |
| $\tau^2_m$ | Membrane Time Constant (I) | 7.06ms |
| $C_m$ | Membrane Capacitance (Default) | $3.96 \times 10^6$pF |
| $C^1_m$ | Membrane Capacitance (HGM, LGM, I) | $0.8 \times 10^6$pF |

The Allostatic Controller of Persistent States architecture (AlloNet) starts with a group of neurons that function as afferent neurons, providing input for the downstream tasks to be done by the external perception neuron (figure 1A). These inputs are used by the Hammel model to self-regulate internal beliefs, representing the interaction with the environment. At its core, we implement a spatially structured network, i.e. the ring attractor network, with $N$ neurons that are fine-tuned to maintain persistent states or bumps. These persistent states are defined to be the internal beliefs that will be controlled by the Hammel model. Each neuron in the model is an Integrate and fire (IAF) neuron, and all the synapses are based on alpha synapses (Gerstner et al. (2014)) with decay constant $\tau^k_{ex}$, where $k$ varies based on the specific sub-networks.

The synaptic weights of these neurons are defined by a symmetry of local excitation and surrounding inhibition to ensure the stability of the persistent states. These synaptic weights, $w$, can be defined by the following equation:

$$w_{ij} = \frac{\sigma_2 e^{-\frac{d_{ij}{}^2}{2\sigma_1^2}} - \sigma_1 e^{-\frac{d_{ij}{}^2}{2\sigma_2^2}}}{\sigma_2 - \sigma_1} \tag{1}$$

Here, $i$ represents the pre-synaptic neuron and $j$ represents the post-synaptic neuron. The terms $e^{-\frac{d_{ij}{}^2}{2\sigma_1^2}}$ and $e^{-\frac{d_{ij}{}^2}{2\sigma_2^2}}$ are Gaussian functions that define the spread of the inhibition and excitation within the network, where the standard deviation $\sigma_2$ and $\sigma_1$ act as parameters controlling the width of the two Gaussian functions. The parameters $d_{ij}$ here is a distance-dependent coupling between the pre-synaptic and post-synaptic neuron, where we can consider as $d(|i-j|)$. To scale the weights for larger or smaller network populations, we define $\sigma_n = \frac{\delta_n 100}{N}$, where $n \in \{1, 2\}$ and $N$ is the population size. The model used in our experiment consists of (N = 100) neurons with $\sigma_2 = 5$ and $\sigma_1 = 10$ being used for tuning the parameters of the network. Thus, the $\delta_n$ parameter is the standard deviation of the Gaussian function that will be scaled based on this population and standard deviation parameters. The tuned parameters of the ring attractor network are provided in table 1.

To control the position, we use a homeostatic circuit inspired by the Hammel model of autonomic nervous system for temperature regulation in mammals (Boulant (2006)). This model is composed of four neurons (figure 1B). In the original model, an insensitive neuron (I) receives input for the model and a sensitive neuron (S) function as a set-regulatory point. The excitation-inhibition balance

of these two neurons is processed through two downstream neurons. These neurons control opposite effectors that aim to either generate or lose heat. This depends on a reference signal relayed to some sensitive neurons by heat receptors in the skin or internal tissues. We adapt this concept to align internal representations, i.e. the localized bump activity representing the system's predicted belief, toward the external inputs in our model. In our adaptation of the model, the reference signal that functions similarly to the insensitive neuron corresponds towards system's predicted belief, and the sensitive neuron that are typically a set-regulatory point can be dynamically adjusted to responses from the environment. This meant that the effectors are the system's belief requiring a mechanism to shift its perception. Thus, we introduce a method that would allow the Hammel Model to shift the localized activity of the ring attractor and be fed back into the network to form a closed control loop.

It is known that under a symmetric kernel with local excitation and surrounding inhibition, the ring attractor network can hold persistent activities in a localized region (Gerstner et al. (2014)). These activities can be directed in a particular direction within the ring attractor by breaking the symmetry of activities in the network. Typically, these are achieved through offsets implemented by an asymmetric connection (Khona & Fiete (2022); Zhang (1996)). Thus, we hypothesize that this control on shifting the position of the bump can be maintained by breaking the symmetry through gain modulation, which will be explained in the next section. This mechanism is in charge of an outer layer of two population of neurons connected to the ring attractor as shown in figure (1C). The homeostatic effector neurons of the Hammel model are connected to the Left Gain Modulation (LGM) and (RGM) neurons to simulate the process of generating or losing heat, respectively.

Finally, a readout network is defined to feed back into the homeostasis model as a reference signal, where these inputs are based upon the position of the bump in the ring attractor (figure 1D). This circuit is inspired from models of familiarity discrimination that occur in the Perirhinal Cortex for processing information as a downstream task (Bogacz R (2001)). This model is used for creating a reference signal for representation perception neurons (figure 1B) from a cumulation of ring attractor neurons' responses. This usually involves three layer of neurons: representation neurons, implemented by the neurons in the ring attractor, familiarity discrimination neurons, referred to as the number discrimination neurons in our model; and decision neurons, implemented by the representation perception neuron in the Hammel Model (sensitive neurons) (figure 1D). These decision neurons project downstream specific rate encodings based on representations from the ring attractor. The external output that drives the changing threshold in the Hammel model as a function of the input works as an accumulation of responses from the environment (figure 1D). A use case for these mechanisms in our model will be presented later.

## 2 CONTROLLING THE BUMP

The allostatic control of the bump in the ring attractor is achieved in two stages, which can be fine-tuned depending on the application. In the first stage, we determine the parameters of the homeostasis model such that the effector neurons vary (almost) linearly as function of the difference in input rate between sensitive and insensitive neurons (figure 2A). In this figure, we have isolated the homeostasis model and replaced the inputs with $I_{syn} = I_{poisson} + I_{bg}$, where $I_{poisson}$ represents the simulated Poisson spike train with a given rate, and $I_{bg}$ is the background noise. This background noise helps linearize the firing rate of inputs to the homeostasis model.

In the second stage, we tune the gain modulation neurons so that they can shift the position of the bump (figure 2B). To achieve this, we first note that the architecture shown in figure 1C implies that the input to the $i$th LGM neuron has two terms:

$$I_{syn}^i = I_{ring}^{i-1} + I_{HL}, \tag{2}$$

where $I_{HL}$ is the output from the "Low" effector of the homeostasis model, and $I_{ring}^{i-1}$ is the output from the ring attractor. Similarly, the input $i$th HGM neuron is given by

$$I_{syn}^i = I_{ring}^{i+1} + I_{HH}, \tag{3}$$

where $I_{HH}$ is the output from the "High" effector of the homeostasis model. Since $i$ represents the neuron index of $N$ in the ring attractor, we consider the $i$ for $I_{HL}$ and $IHH$ to their respective positions in the ring attractor.

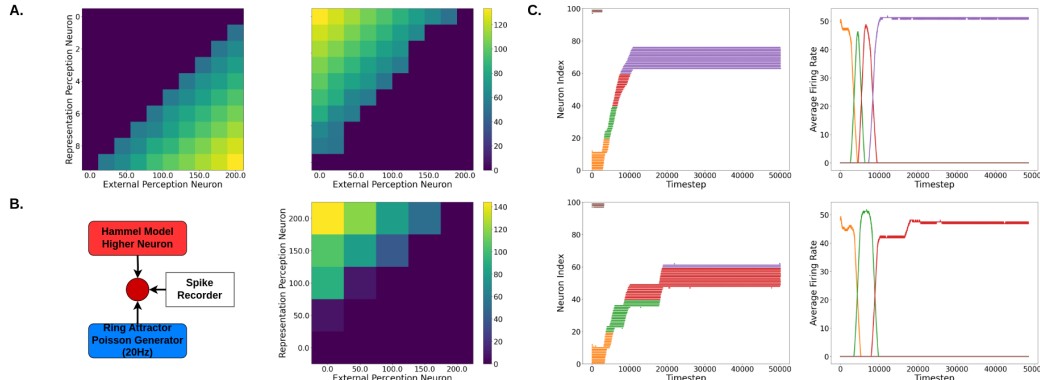

Figure 2: Overview the fine-tuning stages for the model. Each time-step used by the simulator is equivalent to an interval of 1ms. **A**. Heat map for the firing rate in ratio to the two inputs to the Hammel model, simulated by a Poisson generator. Left. Activity (Firing rate) in lower neuron for different input frequencies. Right. Activity of higher neuron. The model is simulated for 10000 time-steps for each combination of input frequencies. **B.** Setup for the fine-tuning process of the gain modulation neurons as described in the text. A similar setup from diagram A. Poisson generators are then used for simulating the bump of the ring attractor and the Hammel model effector output. The colors represent firing rates in response to different combinations of firing. **C.** Response of the bump for two different frequencies of the external input (50000 time-steps). Left. Raster plot. Right. Average firing rate of the readout neurons. The colors represent the ranges for different numbers (see below).

Each of these neurons, in turn, inhibits the next (or previous) ring attractor neuron. Consequently, whenever the ring and the Hammel effector neurons are co-active, they will unbalance the activity in the ring attractor. Moreover, we need to guarantee that when only one of the two (either the neuron in the ring attractor or the corresponding output of the homeostasis model) is active, the LGM or HGM neuron is close to the threshold and do not fire:

$$V_m(I_{ring}^{i\pm1}) \approx V_m(I_{HH}) = V_{th} - h < V(I_{ring}^{i\pm1} + I_{HH}), \tag{4}$$

where $V_m$ is the membrane potential of the LGM or HGM neurons, $V_{th}$ is the firing threshold and $h$ is a small arbitrary constant (figure 2B). With these considerations, we can move the bump so that it matches internal and external representations (figure 2C). All the simulations are performed using NEST simulator (Gewaltig & Diesmann (2007)) in a standard Laptop computer (13th Gen Intel® Core™ i7-13700HX, 16GB Ram, Nvidia RTX4060). The parameters of the Hammel and Gain modulation networks are given in table 1.

## 3 SUBITIZING AS AN ALLOSTATIC PROCESS: BEHAVIORAL RESPONSES

As an example, we introduce the AlloNet as a model of subitizing. Subitizing can be seen as a "number sense" that we define, in this paper, as an allostatic match between external inputs and internal representations (figure 3A).

To set up our model to subitize, we define the ring attractor as a putative number line representation (Hamdan & Gunderson (2017); Revkin (2008)). The input to the network is represented as four Poisson spike train generators which feeds onto the insensitive neurons (here renamed as external perception neurons) and tuned to generate firing rates in the range $[0 - 200]$Hz depending on the number of co-activated inputs. Note that this guarantees permutation invariance of the inputs (i.e., activation sequences $\{1, 0, 1, 0\}$ generate the same input as $\{0, 1, 0, 1\}$ and so on).

We define the number representations by the bijection:

$$\{[0Hz, 50Hz], [51Hz, 100Hz], [101Hz, 150Hz], [151Hz, 200Hz]\} \mapsto \{1, 2, 3, 4\}. \tag{5}$$

As seen in figure 2C, once the input is given, the model adjusts its internal representation to match to the corresponding numerosity. Similarly, we set up four readout neurons to match the corresponding frequencies as those of the input.

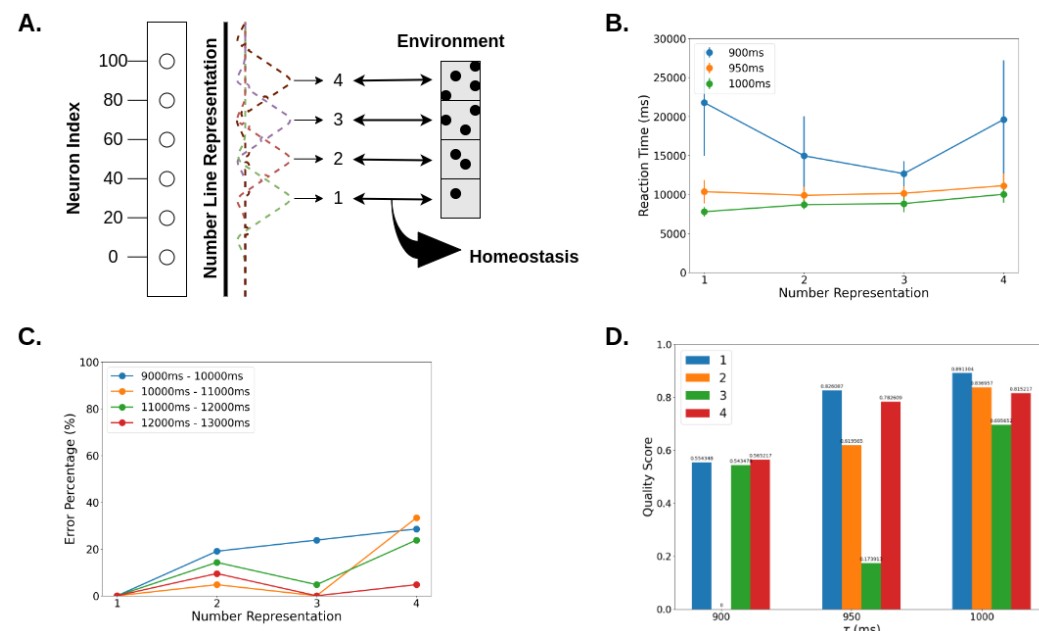

Figure 3: Simulated behavioral responses for a subitizing task. A. We use the AlloNet model to define a homeostatic match between the internal representation of numbers and stimuli in the environment (dots). B. Reaction times in association to number representation defined as the time of first spike in the correct representation. Responses in S2 neurons are used for determining reaction time. The experiments are repeated for different synaptic time constants of the HGM and LGM. C. Error percentage based on the responses at different stopping time-steps in 20 trials (see text) **D.** Bar plot to define quality score for different excitatory synaptic time constants and different numbers. The different bar represents the number being represented (color-coded) and the groupings of these bars are reported based on their synaptic time constants.

### 3.1 THE ALLONET MODEL COMPARISON TO BEHAVIORALLY PLAUSIBLE REACTIONS TIMES AND ERROR RATES

To investigate the relevance of our model as a model of subitizing, we simulated two common subitizing experiments in humans. In the first experiment, we tested whether the network reproduces relevant reaction times as those observed in humans. To do so, we define reaction time as the time of the first spike from the readout neuron corresponding to the input numerosity. The results are shown in figure 3B. We parameterize the experiment further by changing the excitatory synaptic time constant of the gain modulation network.

The model reproduces behavioral responses in two significant ways and fails in one crucial aspect. For longer synaptic time constants (1000ms), the reaction time increases as a function of numerosity and also becomes more variable. However, with shorter synaptic time constants (900ms), the model presents paradoxical behavior: smaller numerosities (numbers less than 3) are noticeably slower than higher numerosities, while this behavior reverses for numerosities higher than 3. In the case of human data, it takes less than half a second to perceive the presence of one, two, or three objects, but the speed and accuracy fall dramatically beyond this limit (Dehaene (2011)). This human data showcases two aspects of the model, where longer synaptic time constants (1000ms) aligned to human responses for numbers less than 3, while short synaptic time constants (900ms) aligned with the sudden increase in reaction time for humans on numbers greater than 3. However, the model fails on the aspects of the dynamics being much slower than human reaction times.

An additional experiment evaluated the error rate of the model (figure 3c). To do so, we fixed a specific time during the run iteration in which the model should report the perceived numerosity and tested the error rate for four different ranges of response timesteps. Here, we define error rate as (number of misfired timestep ÷ total timestep). For early response times, error rate increases with

numerosity as observed in experiments (Dehaene & Cohen (1994)). However, for longer waiting times, responses become inconsistent. We investigate this in the following section.

Lastly, we design an experiment to evaluate the performance of the ring attractor network by analyzing the representation quality over time within defined spatial intervals. Specifically, this analysis investigates a measure on reaching the desired state quickly while remaining stable within that predefined desired region. To calculate the quality score, we use the centroid calculation of the network, $P(t)$. These are then classified as right or wrong time based on whether the centroid is in the desired region of bump activity. Thus, we define the quality score as follows:

$$\text{Quality} = \frac{\text{Number of right times}}{\text{Number of right times} + \text{Number of wrong times}} \tag{6}$$

The analysis shows that longer synaptic time constant performs better. Thus, we will discuss how different synaptic time constants affect the persistent states in the following section.

## 4 SUBITIZING AS AN ALLOSTATIC PROCESS: NEURAL DYNAMICS

To understand the responses for different excitatory synaptic time constants, we investigate through a heat-map that visualizes the activity of neurons across different time windows (figure 4). The excitatory synaptic time constant may affect the reaction time of the network due to two possible outcomes: the slow angular speed of the shifting cues in the ring attractor network or instability of persistent states in the network. Although the first outcome can be directly inferred as time taken to be inversely proportional to the angular speed (Seen in $\tau = 900$ in comparison to $\tau = 1000$ (figure 4)), the second outcome deters that high angular speed can also affect the network to have slow response time. In the case that the network has excessive angular speed (greater than 0.018 rad/ms), this may diminish the ring attractor's capability to track towards the desired persistent state and oscillate between different signals (Chen (2024)) (Seen in $\tau = 1250$ in comparison to $\tau = 1000$ (figure 4)).

As a further investigation of the network, we start by studying the responses of a single neuron to different numerosities over different trials. As an example, we show neuron number 63 in the ring attractor which is tuned to respond to number 3 (figure 3A).

The response of a given neuron is composed of transients for higher numerosities (as the bump transits its neighborhood) and persistent activity for the numerosity it is tuned to. Note however that for later times the bump becomes unstable and starts a new wander due to the noise of in the network. This explains the inconsistent error rates for higher response times in figure 3C.

We also studied the gain field in the vicinity of neuron 63, composed of only transient responses for numerosities greater than or equal to the tuned numerosity (figure 5A). These two profiles of activity resemble the responses of the Entorhinal Cortex, the Parahipocampal Cortex and the Hippocampus in humans (Kutter et al. (2023)).

To understand a bit better the dynamics of the bump, we studied the position coordinate (Hopfield (2015)) of the bump and its speed. The position coordinate consistently represents the numerosity across trials (figure 5B) and the velocity shows evidence of a constant acceleration towards the corresponding numerosity. Furthermore, by investigating the scaling of the speed with the synaptic time constant we discover the reason for the difference in reaction times for slower time constants: the bump movement profile becomes shallower for lower numerosities due to lack of excitation from the input (figure 5C, dotted line).

## 5 CONCLUSIONS AND FUTURE WORK

We have proposed a model of allostatic control of persistent states in spiking neural networks, named the AlloNet. Additionally, we propose it as a computation model of subitizing in animals, defining the number sense as a homeostatic alignment of internal representations and external inputs. Our model reproduces qualitatively important behavioral aspects of subitizing in humans as well as observed neural dynamics of the regions known to be involved in number perception.

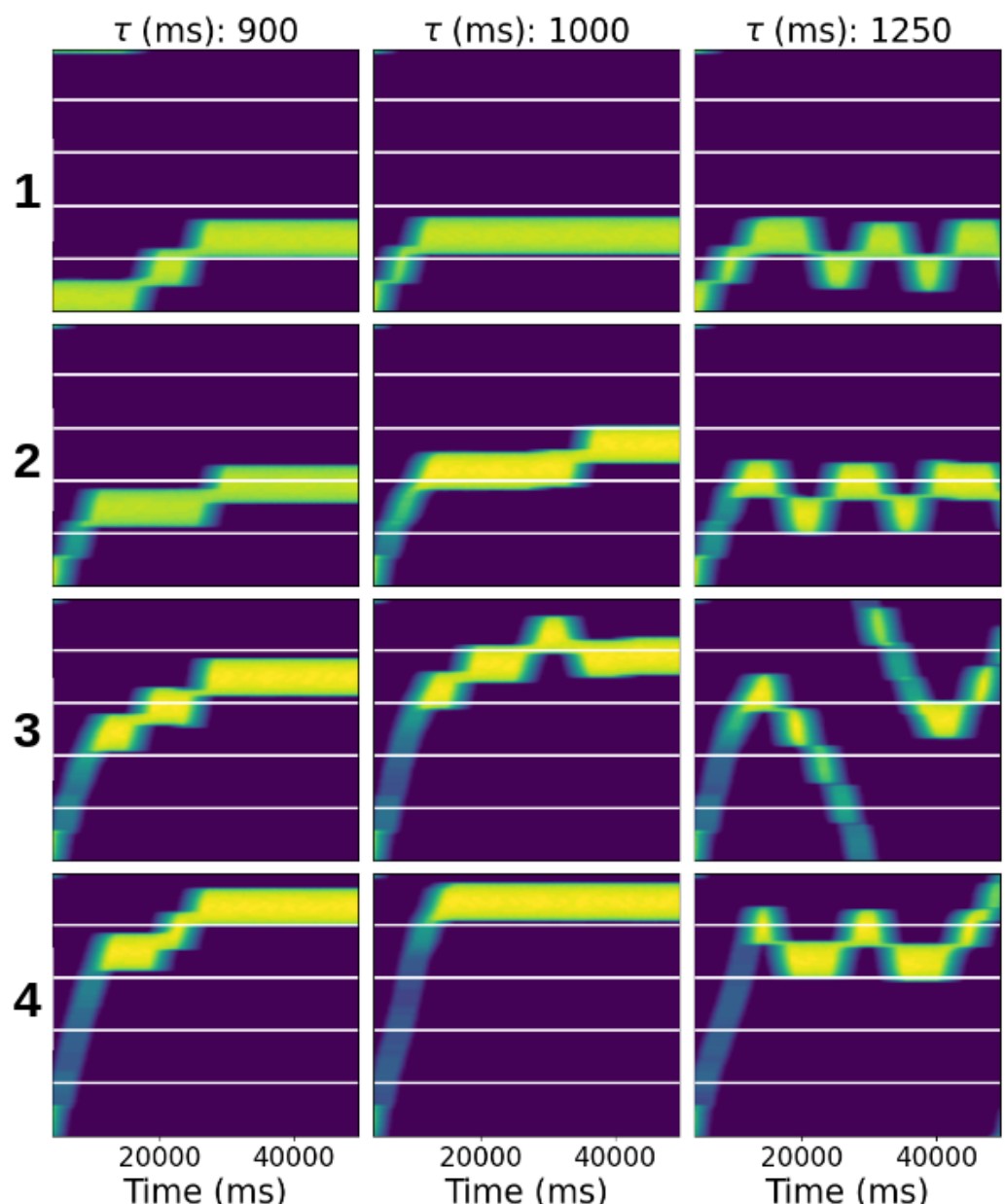

Figure 4: Heat-map to plot the network persistent state over time. The y-axis of the plot is the number representation index, and the y-axis of the sub-plot is the neuron index of the ring attractor network. The x-axis of the plot is the column for different synaptic time constant runs and the x-axis of the sub-plot is the time of the run iteration. The white lines in the sub-plots define the different numerosities, i.e., [starting point,1,2,3,4] from bottom to top.

The model is able "track" a static property of the environment by dynamically adjusting an space-like internal model to match a function of the input which, due to the simplicity of the input network and the properties of neural integration, is numerosity. That is, this model follows the premise of perception as active process (Parr et al. (2022)), moreover, that perception is self-regulatory in that the internal and external models need to find a match through the lens of a shared interface (Parr et al. (2022)). This interface is instantiated in the Hammel model.

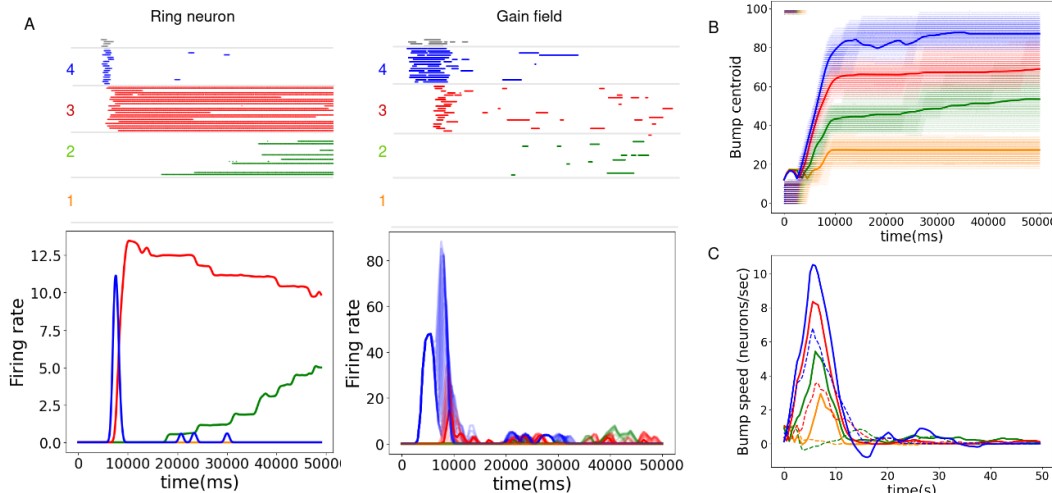

Figure 5: Neural dynamics and dynamics of the bump. The color is encoded to the location of the bump activity (shown by the number represented on the left of the plots in figure 5A) A. Responses of a single neuron of the ring attractor (63) sensitive to number 3. Top. Raster plot for 20 trials of each numerosity. Bottom. Corresponding firing rates. Right. Responses of the gain field in the vicinity of the ring ($\pm 10$ neurons). B. Centroid of the bump computed as the expected value of position (neuron) with respect to the normalized firing rate. The spike trains are also plotted in the background for each numerosity. C. Speed of the bump smoothed with 5 samples sliding window for each numerosity. Dotted line is the corresponding step for lower synaptic time constant ($\tau_{ex} = 900ms$).

The Hammel model is involved in self-regulation. It works by minimizing the error between a given input and a fixed set value. By setting variability and abstraction to the threshold (as numerosity is not a one-dimensional signal), we allowed the model to handle more general representations. We think that this idea can be extended further to solve more general computational problems (Hopfield (2015)).

The self-regulatory aspect of the model implies that there is an autonomic response (internal or covert change) and a behavioral response (external or cover). This is analogous to the duality between subitizing or approximating, counting (Dehaene & Cohen (1994)), both involve different systems and one of them is behavioral (Pecyna et al. (2022)) while the latter is purely autonomic or spontaneous (Castaldi et al. (2021)) and is modality independent (Togoli & Arrighi (2021)).

The self-regulatory aspect of subitizing is thus the main contribution of this work. This change of perspective has the potential to unify different systems that share the same properties of the current model under a common view that could be considered as a neuromorphic version of active inference (Parr et al. (2022)). Indeed, similar models have been described in insects for controlling head direction (Pisokas et al. (2020)) which have inspired computational (Stentiford et al. (2024)) and robotic models (Robinson et al. (2022)). However, in contrast with many existing models, our current model does not include any sort of learning but adaptation, a property that is key not only for subitizing (Togoli & Arrighi (2021)) but for general perception and control as well. We think learning will add degrees of freedom to the model like the possibility of building a full Approximate Number System.

There are still many challenges to overcome, and they are the focus of current computational and theoretical work. One of them is the fact that the model has to be slow to be stable (i.e. the difference between the fast time scales of spiking and the slow dynamics of the bump) needs to be significant. Furthermore, the qualitative descriptions here need to be validated against actual data that not only approaches the difference in age when subitizing but, perhaps more importantly, the dysfunction of such systems. We think our overall approach is productive in investigating numbers and similar abstract senses in animals as well as in robots.

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
