# OpenReview forum: "Allostatic Control of Persistent States in Spiking Neural Networks for Perception and Computation"
_ICLR.cc/2025/Conference — Submitted to ICLR 2025_

### Official Review · Reviewer_BvKC · 2024-10-30

**Soundness:** 2
**Presentation:** 2
**Contribution:** 2
**Rating:** 3
**Confidence:** 4

**Summary:**

This work develops a framework that dynamically adjusts representation based on environmental inputs, drawing inspiration from the body’s temperature regulation system and incorporating the concept of allostasis. To encode stimuli, the model employs a ring attractor network, while an allostasis module called “Hammel model” serves to compare internal and external variables. This module then outputs gain control signals to drive the ring attractor’s response moving. The model was verified by a subitizing task.

**Strengths:**

1.	The introduction section provides a comprehensive overview of both numerosity and attractor networks.
2.	The integration of physiological concepts into neural circuit modeling is innovative and inspiring.

**Weaknesses:**

1.	Improper Citations:  For instance, in lines 173-174, citing Zhang, K. (1996) is essential for discussing the asymmetry in continuous attractor connections. Additionally, it appears that Fig. 1c is likely influenced by Fig. 1d in Wilson, R.I. (2023), but lacks appropriate citation.
2.	Lack of Clarity on Network Dynamics and Analysis: The model does not provide a clear definition of network dynamics or a detailed theoretical analysis of the results.
3.	Biologically Implausible Model Setup: The model’s structure and parameters are set in bio-unplausible ways. Specifically:
	- The synaptic time constants far exceed the plausible range, making the comparison across time constants in Figs. 4 and 5 less meaningful.
	- LGM and HGM neurons were designated as inhibitory. I agree that it is a feasible way to drive the ring attractor, but there has been both experimental and theoretical evidence showing that P-EN neurons in Drosophila brains fulfill this role as excitatory neurons (see Zhang, W., Wu, Y. N., & Wu, S., 2022; Mussells Pires, P., Zhang, L., Parache, V., Abbott, L. F., & Maimon, G., 2024).
4.	Failure to Model Human Behavior Accurately: Although section 3.1 attempts a loose comparison between the model and human behavior, the model does not successfully replicate human behavioral patterns.
5.	Unpolished text and figures: The text contains several typos, unified terminologies and missing punctuation.

**Questions:**

1.	There’s a saying suggesting that the number line is organized not linearly but log-linearly (Testolin, A., & McClelland, J. L., 2021). Could the present model still function effectively if the ring attractor network were replaced with a log-linear number line?
2.	Ideally, each point on the ring attractor should be neutrally stable, meaning the probability of bump initiation would be equal at any point on the ring. However, the initiation point significantly affects reaction time. How do you ensure the bump consistently initiates from the same point each time?
3.	In Eq. 4, should it be V_th-h so that the potential is below the threshold? Also, could you clarify lines 241-243 preceding Eq. 4? I ask because, in my understanding, the bump should remain stable as long as LGM and HGM neurons are firing equally but not necessarily silent.

---

> ### Author Response · Authors · 2024-12-02
>
> ### Weaknesses
>
> W1. Improper Citations: For instance, in lines 173-174, citing Zhang, K. (1996) is essential for discussing the asymmetry in continuous attractor connections. Additionally, it appears that Fig. 1c is likely influenced by Fig. 1d in Wilson, R.I. (2023), but lacks appropriate citation.
>
> A: Thanks for noticing this. I have just added it. As for the second structure, figure 1C is not influenced by Wilson, R.I. (2023). The figure is more influenced by the usual ring attractor with the inspiration of cue integration (its inhibitory in this case though) (external cues being the inhibitory connection from LGM and HGM).
>
> W2. Lack of Clarity on Network Dynamics and Analysis: The model does not provide a clear definition of network dynamics or a detailed theoretical analysis of the results.
>
> A: Thank you for your comment. A full analysis of the dynamics was not intended in the paper and different aspects of the dynamics of ring attractors can be found elsewhere. What particular aspects of the dynamics do you think should be included? A key one we can identify is the stability of the steady state of the bump. We are working on this at the moment, but we considered out of scope for the current paper.
>
> W3. Biologically Implausible Model Setup: The model’s structure and parameters are set in bio-unplausible ways. Specifically:
> The synaptic time constants far exceed the plausible range, making the comparison across time constants in Figs. 4 and 5 less meaningful.
> LGM and HGM neurons were designated as inhibitory. I agree that it is a feasible way to drive the ring attractor, but there has been both experimental and theoretical evidence showing that P-EN neurons in Drosophila brains fulfill this role as excitatory neurons (see Zhang, W., Wu, Y. N., & Wu, S., 2022; Mussells Pires, P., Zhang, L., Parache, V., Abbott, L. F., & Maimon, G., 2024).
>
> A: Thank you for pointing this out. Current iteration of the work initially started from C. R. Laing et al. (2001) where we got our initial hyperparameters after which we optimised towards finding a suitable parameter for our work. The main exploration we did is on capacitance, membrane time constant, input current and the connection weights. I agree that this seems biologically implausible at this stage. As for the second part, there have been models that mainly use excitatory connections for cue integration, evidence on using inhibitory connections have been shown as well. E.g. Chang et al. (2023) (Global inhibition in head‑direction neural circuits: a systematic comparison between connectome‑based spiking neural circuit models) showcases this as a global inhibition on the ring attractor but in our case we specify the inhibition towards a specific region to cause drift in the network.
>
> W4. Failure to Model Human Behavior Accurately: Although section 3.1 attempts a loose comparison between the model and human behavior, the model does not successfully replicate human behavioral patterns.
>
> A: Thank you for pointing this out. As you have said, the model fails to induce similar responses to that of human behavior and the limitations for this have been explained in section 4 (reasons for why this network reacts slower). In this work we have provided a plausible architecture to implement a number line representation with spiking neural network and a potential mechanism for subitizing. This was not necessarily based on any measurement, but on the functional requirements that such circuit must satisfy. For future work, we plan to tackle the latency problem through the introduction of a homeostatic plasticity to maintain a balance of spiking inputs towards the network which should allow us to ensure a faster response (due to better control over the drift process).
>
> W5. Unpolished text and figures: The text contains several typos, unified terminologies and missing punctuation.
>
> A: Thank you for pointing this out. The manuscript has been revised to address these issues.
>
> ### Questions
>
> Q1. There’s a saying suggesting that the number line is organized not linearly but log-linearly (Testolin, A., & McClelland, J. L., 2021). Could the present model still function effectively if the ring attractor network were replaced with a log-linear number line?
>
> A: The model is meant mainly for a linear number line (for small numerosities). We consider these two as separate systems where the mechanisms considered for a log-linear number line would be different (for large numerosities). This is in line with what Leslie et al 2008 proposed. Interestingly, the log-linear behaviour could emerge in our model from the dynamical properties of the bump, as it takes some time to reach/match the perceived numerosity; this time could be made logarithmic.

---

> > ### Author Response · Authors · 2024-12-02
> >
> > Q2. Ideally, each point on the ring attractor should be neutrally stable, meaning the probability of bump initiation would be equal at any point on the ring. However, the initiation point significantly affects reaction time. How do you ensure the bump consistently initiates from the same point each time?
> >
> > A: This is not the intended use for the ring attractor of our model. It is supposed to work as a number line, therefore, only the point corresponding to zero will allow the initiation of bump activity. The rest of the neurons will only be excited by the propagation of the bump. So, not every point in the ring has the same probability of initiation.
> >
> > Q3. In Eq. 4, should it be V_th-h so that the potential is below the threshold? Also, could you clarify lines 241-243 preceding Eq. 4? I ask because, in my understanding, the bump should remain stable as long as LGM and HGM neurons are firing equally but not necessarily silent.
> >
> > A: Thanks for pointing this out. You are correct, we have corrected this typo.. What we wanted to convey was on how the LGM and HGM behaves here. The ring attractor is not necessarily silent. What it means is that on regions of bump activity (where there is spiking activity specifically) it would balance with the LGM and HGM (if they are active) to perturb upon the surrounding region of the bump to create instability that should create drift on the network. In summary, we want the LGM and HGM neurons to be able to break the symmetry of the synaptic kernel under small perturbations.
> >
> > ### References
> >
> > Chang, N., Huang, HP. & Lo, CC. Global inhibition in head-direction neural circuits: a systematic comparison between connectome-based spiking neural circuit models. J Comp Physiol A 209, 721–735 (2023). https://doi.org/10.1007/s00359-023-01615-z
> >
> > Laing, C. R., & Chow, C. C. (2001). Stationary bumps in networks of spiking neurons. Neural Computation, 13(7), 1473–1494.

---

### Official Review · Reviewer_eoxw · 2024-11-04

**Soundness:** 2
**Presentation:** 2
**Contribution:** 2
**Rating:** 3
**Confidence:** 4

**Summary:**

The paper investigates how structured networks, like ring attractors, control bump activity in response to external stimuli? The authors propose a network model combining a Hammel model and ring attractor network, with high/low gain modulation neurons driving the bump shift to align with an external stimulus. They argue that this model, due to its control over bump positioning, could model subitization (rapid counting) and compare it to biological data.

**Strengths:**

1. The authors propose a novel network model based on the ring attractor, enhanced with high/low gain modulation neurons to manage bump shifts, with the Hammel model controlling these shifts. This enables flexible positioning of the bump to match external signals.
2. Some aspects of the model’s performance on counting tasks align with human data, particularly when varying the synaptic time constant.

**Weaknesses:**

1. The counting task demonstration is very simple; the numerical information is directly encoded in the external signal’s firing rate.
2. Although certain model properties align with human data, they do so under varying synaptic time constants, which are not well explained. Additionally, the model’s reaction times are significantly slower than human responses, by about an order of magnitude.
3. The necessity of using a spiking neural network is not adequately justified.
4. The model’s structure, specifically the gain modulation neurons and the Hammel model component, lacks biological explanation.
5. The network dynamics and mechanisms are underexplained. For instance, in the connection weight formula $w_{ij}$, the term $d_{ij}$ is undefined, though it appears to represent the distance between neurons in the ring.


### Mino
1. In Equation 1, $d_{ij}$ is undefined.
2. Figure 1, with its four subplots, could be clearer if organized differently.
3. In Figure 2, specify units for the x-axis (probably ms).
4. The synaptic decay constants \tau used are unusually long compared to standard neuron models. More explanation of these values and their role would improve readability.

**Questions:**

1. How did you decide on this setup, specifically the use of high/low gain modulation neurons to drive shifts and the Hammel model to control them? Is there biological evidence supporting these choices?
2. Could a similar task be achieved with a rate model? If so, how would its reaction time compare to that of the spiking model?
3. The alpha synapse decay constant for HGM and LGM is approximately 1000ms, which is unusually long for neurons. Could you provide more reasoning behind this choice?
4. The input setup is unclear. Do the four Poisson spike generators fire at the same rate? If not, how do they differ, and why is numerical information encoded in firing rates? Is there experimental evidence for this encoding, or could a different encoding method (like one-hot vectors or embeddings) be used?

---

> ### Author Response · Authors · 2024-12-02
>
> ### Weaknesses
>
> W1. The counting task demonstration is very simple; the numerical information is directly encoded in the external signal’s firing rate.
>
> A: Our initial experiment presented here is to capture the dynamics of a linear number line with subitization being the key here. The experiment is simple but not trivial . Considering Leslie et al (2008)’s association of linear number line to visual-spatial skills, we intend to extend this work towards robotic manipulation experiments for further studies (attributing to the capability of the network being able to do hand-eye coordination tasks).
>
> W2. Although certain model properties align with human data, they do so under varying synaptic time constants, which are not well explained. Additionally, the model’s reaction times are significantly slower than human responses, by about an order of magnitude.
>
> A: Thank you for your comment. The synaptic constants presented in this paper were found experimentally in the model while searching for a regime that allowed for almost linear velocities in response to the hammel input. We found that the synaptic constant is a key parameter to change the behavior of the model.
>
> W3. The necessity of using a spiking neural network is not adequately justified.
>
> A: The key part of the model is the gain modulation provided and it relies heavily on the timing based spikes provided by the homeostasis network (Hammel) and the ring attractor network. Thus, spiking neural networks is essential to this network.
>
> W4. The model’s structure, specifically the gain modulation neurons and the Hammel model component, lacks biological explanation.
>
> A: We tried to find micro-circuit motifs that could be used to achieve the target representation of the number line. In doing so, we expect to generate some hypotheses about how these circuits could be organised in different animals. We agree we need to do a better work with comparing existing neural circuits that have been mapped.
>
> W5. The network dynamics and mechanisms are underexplained. For instance, in the connection weight formula, w_ij the term d_ij  is undefined, though it appears to represent the distance between neurons in the ring.
>
> A: Thank you for pointing this out. To address this, we have added the following explanation on the formula: The parameters $d_{ij}$ here is a distance-dependent coupling between the pre-synaptic and post-synaptic neuron, where we can consider as $d(|i-j|)$.
>
> Minor
>
> W1. In Equation 1, d_ij is undefined.
>
> A: This has been addressed as per major weaknesses 5.
>
> W2. Figure 1, with its four subplots, could be clearer if organized differently.
>
> A: Thank you for pointing out an issue with clarity in figure 1. To address this problem, I have adjusted changes to it as follows: Fig 1A has been increased in size to have better clarity. It is one of the important parts of the architecture so we have increased the size taken up by it.
> For all figures, the size of text has been increased as well.
>
> W3. In Figure 2, specify units for the x-axis (probably ms).
>
> A: The following part indicates the time-step taken by the simulator itself. To address this, I have put in that 1 time-step in the simulator is equivalent to 1ms (according to parameters used in nest simulator).
>
> W4. The synaptic decay constants \tau used are unusually long compared to standard neuron models. More explanation of these values and their role would improve readability.
>
> A: Thank you for this suggestion. The longer decay constants are the product of the design process of the circuit. We did a parameter exploration, as presented in the paper, and those constants are the ones that allowed for almost linear velocities in the bump. In short, they are a design feature.  In line with this suggestion, I have added it to table 1 to clarify this:  “The following parameters are initially based on (\cite{Laing2001}), where a parameter search is done to find the regime for the ring attractor drift velocity to be linear to the Hammel input.”
>
> ### Questions
>
> Q1. How did you decide on this setup, specifically the use of high/low gain modulation neurons to drive shifts and the Hammel model to control them? Is there biological evidence supporting these choices?
>
> A: This is too, to some extent, a design feature. We looked for ways of making the synaptic kernel asymmetric dynamically. We wanted to avoid synaptic plasticity because our target application down the line is neuromorphic hardware (in particular Spinnaker) in which plasticity can be complicated to implement.
>
> Q2. Could a similar task be achieved with a rate model? If so, how would its reaction time compare to that of the spiking model?
>
> A: Thanks for this comment. This is an interesting comparison; in principle, spatially structure networks have very consistent mean field representations, so we expect the behaviour to be similar. The benefit of our current implementation is that it can be run on neuromorphic hardware.

---

> > ### Author Response · Authors · 2024-12-02
> >
> > Q3. The alpha synapse decay constant for HGM and LGM is approximately 1000ms, which is unusually long for neurons. Could you provide more reasoning behind this choice?
> >
> > A: Initial parameters were chosen from C. R. Laing (2001) to sustain a plausible parameter for sustaining the bump in the ring attractor. Different sub-regions (defined in Figure 1) are then used for a parameter search to find a regime that allows for almost linear velocities in response to the hammel input. This has been added to Table 1 for clarification with the following line: “The following parameters are initially based on (\cite{Laing2001}), where a parameter search is done to find the regime for the ring attractor drift velocity to be linear to the Hammel input.”
> >
> > Q4. The input setup is unclear. Do the four Poisson spike generators fire at the same rate? If not, how do they differ, and why is numerical information encoded in firing rates? Is there experimental evidence for this encoding, or could a different encoding method (like one-hot vectors or embeddings) be used?
> >
> > A: Thank you for your comment. The rate of the four Poisson depends upon the particular numerosity pattern, but it has only two levels depending upon whether the given point in the pattern is present or not. We chose the simplest encoding possible with spiking inputs.
> >
> > ### References
> >
> > Laing, C. R., & Chow, C. C. (2001). Stationary bumps in networks of spiking neurons. Neural Computation, 13(7), 1473–1494.
> >
> > Leslie, Alan M. ; Gelman, Rochel & Gallistel, C. R. (2008). The generative basis of natural number concepts. Trends in Cognitive Sciences 12 (6):213-218.

---

### Official Review · Reviewer_EBGP · 2024-11-06

**Soundness:** 1
**Presentation:** 1
**Contribution:** 2
**Rating:** 3
**Confidence:** 3

**Summary:**

This paper proposes a mechanism to control the activity of spiking neural networks through allostasis, in order to perform a subitising task. The proposed architecture uses the Hammel Model to regulate internal beliefs associated with the task in response to stimuli, and this is used to control bump activity on a ring attractor that encodes numerosity. The authors compare observed reaction times from their model to those of humans performing the same task, and are able to reproduce certain human behavioural responses with their model. They also assess the impact of the value of the neurons' time constant on task performance, and compare a single neurons' activity profile to known neural dynamics for subitising in humans.

**Strengths:**

* The proposed model is original to my knowledge, and is fairly simple to understand.
* The authors have compared observations from their models such as reaction times to those of humans performing the same task.
* A testable prediction, i.e., that small numerosities are encoded using a magnitude-based system (ring attractor in this case, which is central to the model), could potentially be validated using experiments with animals/humans.

**Weaknesses:**

* While the authors are able to recapitulate some observations in humans using their model, i.e., that reaction times increase with numerosity (for a specific value of the time constant), the actual reactions times from the model are far greater than human reaction times (almost 10000 ms vs less than 500 ms) ([Dehaene, 2011](https://psycnet.apa.org/record/2011-10610-000); [Kutter et al., 2023](https://www.nature.com/articles/s41562-023-01709-3)). Do the authors have an explanation for why this is the case, and could it be resolved with better choices for hyperparameters?
* I'm not entirely convinced with the authors' argument that they are able to properly reproduce reaction time patterns. The sharp jump in reaction time for a numerosity of 4 is only observed with a different, faster time constant value of 900ms, while the gradual increase in reaction times with numerosity is only observed with a slower time constant of 1000ms. Furthermore, in the 900ms case, the reaction times for lower numerosities do not match the human data at all, as the authors admit. Could the authors clarify this? I would expect the model to match human observations for a single value of the time constant in order to claim that it reproduces experimental results.
* The model is evaluated only on a single task, i.e., subitising. Especially given the use of a ring attractor and the authors' claims of the model's generalisability, it would be important to evaluate the model on other tasks involving numerosity estimation or tracking a magnitude, such as head direction integration ([Valerio & Taube, 2012](https://www.nature.com/articles/nn.3215)) or average estimation ([Lee & Ma, 2020](https://cognitivesciencesociety.org/cogsci20/papers/0304/0304.pdf)).
* The task is also limited in that it restricts numerosity from 1 to 4. To better compare the observations from the model to previous experiments ([Kutter et al., 2023](https://www.nature.com/articles/s41562-023-01709-3)), the authors should incorporate both small and large numbers (from 0 to 9, for example). If the authors want to align their work better with [Kutter et al. (2023)](https://www.nature.com/articles/s41562-023-01709-3), two different representations could be used for 0-4 vs 5-9 – and in this case it would be important to test whether discrete attractors (fixed points for each of 0-4) match the data better than when using a ring attractor.
* There are other issues with the results and their interpretation. For example, in Fig. 3B, it is not clear how much more variable the reaction times are for higher numerosities with longer time constants (as claimed on line 312). Furthermore, from Fig. 3D, it seems that the "quality score" is very low for numerosity 3 and longer time constants (while for shorter time constants, quality is 0 for numerosity 2 but quality is higher for 3), why is this the case, and is there any experimental evidence of this in animals/humans?
* The authors claim that the neural dynamics of their model are similar to those of several brain regions in human recordings, but this is not substantiated with a metric or even a plot showing qualitative similarity. Furthermore, some of the analysis of neuronal responses has only been done for a single, arbitrary neuron, but it would be important to look at population responses when making comparisons to human data.
* Another weakness is the specificity of these results to hyperparameter and architectural choices. The lack of learning also makes it harder to adapt this model to more complex tasks.
* Finally, the writing lacks clarity and contains grammatical, formatting and some typographical issues. Examples:

  * Line 50 ("can represent to track changes...", unclear what this means), Line 53 ("resemble to the" -> "resemble the"), Line 158-159 ("functions to take input for.."), Line 249 ("in an standard..."), Line 463-465 (sentences are vague, "we allowed to work on"), etc.
  * Several in-text citations are not properly enclosed within brackets.
  * "Excitatiory" -> "Excitatory" (Table 1), inconsistent use of "ise" vs "ize" (British vs American English), "... as a computation model ..." -> "computational", lack of proper spacing before and after parentheses, etc.

  I would encourage the authors to carefully revise the paper to improve its clarity and fix any other grammatical/typographical issues.

**Questions:**

See the Weaknesses section. I have some additional questions:
* I don't entirely understand the use of a ring attractor for this task. For a number line representation, wouldn't a bounded line attractor be a better idea, especially as it does not wrap around from the maximum to the minimum? Furthermore, the use of a continuous attractor makes the representation more akin to estimation rather than categorisation/subitising, and would align better with work such as [Piazza et al. (2002)](https://www.sciencedirect.com/science/article/abs/pii/S1053811901909802?via%3Dihub) claiming that counting and subitising use the same neural circuitry, as opposed to [Kutter et al. (2023)](https://www.nature.com/articles/s41562-023-01709-3). Could the authors clarify this and correct me if I have misunderstood something?
* How do the authors choose the various hyperparameters mentioned in Table 1? Have the authors tested the robustness of the results to changes in hyperparameters apart from the time constant?

---

> ### Author Response · Authors · 2024-12-02
>
> ### Weaknesses
>
> W1. While the authors are able to recapitulate some observations in humans using their model, i.e., that reaction times increase with numerosity (for a specific value of the time constant), the actual reactions times from the model are far greater than human reaction times (almost 10000 ms vs less than 500 ms) (Dehaene, 2011; Kutter et al., 2023). Do the authors have an explanation for why this is the case, and could it be resolved with better choices for hyperparameters?
>
> A: As stated in section 4 there are two reasons for this: a. Slow angular speed b. Too many signals for moving may also make the persistent state oscillate between different signals. To accommodate this, introducing homeostasis plasticity to maintain an ideal speed for drifting the bump would help. This is being considered for future work.
>
> W2. I'm not entirely convinced with the authors' argument that they are able to properly reproduce reaction time patterns. The sharp jump in reaction time for a numerosity of 4 is only observed with a different, faster time constant value of 900ms, while the gradual increase in reaction times with numerosity is only observed with a slower time constant of 1000ms. Furthermore, in the 900ms case, the reaction times for lower numerosities do not match the human data at all, as the authors admit. Could the authors clarify this? I would expect the model to match human observations for a single value of the time constant in order to claim that it reproduces experimental results.
>
> A: A: Thanks for your thoughtful question. We acknowledge that the model's reproduction of reaction time patterns still requires deeper investigation to align more closely with human data.  The unexpectedly large reaction times in our simulations remain an area we are actively exploring. While our current work provides some initial intuitions, as discussed in the paper (e.g., the difference between the first arrival of the bump at the numerosity and the quality of the representation, or how long it stays there), these dynamics introduce additional factors that make direct comparisons with human data complex. We still think that the dynamics of a travelling bump, provide a promising avenue for the understanding of subitizing. Here we summarise some of these reasons:
> The human reaction time data in Figure 1 of Dehaene (1994), which we use for comparison, is based on results from George Mandler et al. (1992) (Figure 7) with a baseline case of 200 ms presentation time. However, the results from our model with synaptic time constants of 900 ms and 1000 ms more closely resemble the behavior observed under the 50 ms presentation time conditions (Figure 4: 50 ms practiced and 50 ms unpracticed). We highlight this not to fully justify the model but to showcase certain qualitative similarities that merit further exploration.
> In our model, the homeostasis model provides different levels of activities based on how much of a numerosity difference is shown to the model. This changes drift speed in a varied manner so what we consider here is the need for this to be in a certain range to ensure that the behavior closely resembles human behavior. Thus, to ensure a better response of perturbation onto the ring attractor, there is a need for considering homeostatic plasticity that relies on the results from both 1000ms and 900ms of our model. This will be done for future work with more on application cases with our current model showing a limited extent of human behavior being shown in this work.
>
> W3. The model is evaluated only on a single task, i.e., subitising. Especially given the use of a ring attractor and the authors' claims of the model's generalisability, it would be important to evaluate the model on other tasks involving numerosity estimation or tracking a magnitude, such as head direction integration (Valerio & Taube, 2012) or average estimation (Lee & Ma, 2020).
>
> A: Our first case of the study is to use the linear number line with relation to spatial skills. For future cases, we are planning to show that this model can introduce visual spatial skills that can be extended towards a robotic use case on hand-eye coordination of robots (robotic manipulation).

---

> > ### Author Response · Authors · 2024-12-02
> >
> > W4. The task is also limited in that it restricts numerosity from 1 to 4. To better compare the observations from the model to previous experiments (Kutter et al., 2023), the authors should incorporate both small and large numbers (from 0 to 9, for example). If the authors want to align their work better with Kutter et al. (2023), two different representations could be used for 0-4 vs 5-9 – and in this case it would be important to test whether discrete attractors (fixed points for each of 0-4) match the data better than when using a ring attractor.
> >
> > A: One inspiration for our model is from Leslie et al. 2008, where we consider small and large numbers to have two different mechanisms. Here, we consider an accumulator model that considers the number line as a linear system. These are more an estimation and hence the reason why we considered mainly on subitization.
> >
> > W5. There are other issues with the results and their interpretation. For example, in Fig. 3B, it is not clear how much more variable the reaction times are for higher numerosities with longer time constants (as claimed on line 312). Furthermore, from Fig. 3D, it seems that the "quality score" is very low for numerosity 3 and longer time constants (while for shorter time constants, quality is 0 for numerosity 2 but quality is higher for 3), why is this the case, and is there any experimental evidence of this in animals/humans?
> >
> > A: Thank you for pointing out this. Yes, unfortunately, these two measures are not always consistent. The reaction times were measured as the time-to-first spike of the readout neurons for each numerosity; while the quality measures how stable is the bump on the ring attractor. It could have happened that you could have a spike transient at the given numerosity while having a very unstable bump (drifting up and down). It is precisely that reason that prompted us to create the quality measure we proposed.
> >
> > W6. The authors claim that the neural dynamics of their model are similar to those of several brain regions in human recordings, but this is not substantiated with a metric or even a plot showing qualitative similarity. Furthermore, some of the analysis of neuronal responses has only been done for a single, arbitrary neuron, but it would be important to look at population responses when making comparisons to human data.
> >
> > A: Thank you for your comment. A full comparison with human data is not within the scope of this paper. But we agree with the reviewer that this is necessary to fully support this claim. Our analyses of the ring are, nevertheless, population analyses (at least regarding the ring). Indeed, scaling the ring up from 100 neurons by orders of magnitude won’t change the qualitative dynamics presented. However, the readout mechanisms and the Hammel circuit are presented at the single neuron level and would be scaled in future work. Regarding the human data, we only mentioned the resemblance of the dynamics observed in the model with those observed in the analyses by Kutter et. al (2023). Indeed, we do believe that a ring attractor (or a spatiotemporal bump) is the most likely explanation of such results.
> >
> > W7. Another weakness is the specificity of these results to hyperparameter and architectural choices. The lack of learning also makes it harder to adapt this model to more complex tasks.
> >
> > A: Thank you for pointing out this weakness in our work. Our current work aims to show how we can use a combination of how to maintain and control representations with future work to be done on how to learn representations defined for this model. Our hyper parameters were initially based on C. R. Laing et al. (2001), where we have done a parameter search to ensure that the drift velocity is almost to responses from the Hammel Input.
> >
> > W8. Finally, the writing lacks clarity and contains grammatical, formatting and some typographical issues. Examples: a. Line 50 ("can represent to track changes...", unclear what this means), Line 53 ("resemble to the" -> "resemble the"), Line 158-159 ("functions to take input for.."), Line 249 ("in an standard..."), Line 463-465 (sentences are vague, "we allowed to work on"), etc. b. Several in-text citations are not properly enclosed within brackets. c. "Excitatiory" -> "Excitatory" (Table 1), inconsistent use of "ise" vs "ize" (British vs American English), "... as a computation model ..." -> "computational", lack of proper spacing before and after parentheses, etc. d. I would encourage the authors to carefully revise the paper to improve its clarity and fix any other grammatical/typographical issues.
> >
> > A: Thank you for this. As for this part, I have double checked and addressed all of the following concerns. I have tried improving clarity on some parts of the wording and fixed the grammatical and typographical errors.

---

> > > ### Author Response · Authors · 2024-12-02
> > >
> > > ### Questions
> > >
> > > Q1. I don't entirely understand the use of a ring attractor for this task. For a number line representation, wouldn't a bounded line attractor be a better idea, especially as it does not wrap around from the maximum to the minimum? Furthermore, the use of a continuous attractor makes the representation more akin to estimation rather than categorisation/subitising, and would align better with work such as Piazza et al. (2002) claiming that counting and subitising use the same neural circuitry, as opposed to Kutter et al. (2023). Could the authors clarify this and correct me if I have misunderstood something?
> > >
> > > A: For the first part of this question, we refer to Sagodi et al (2024). In the case of line attractors for post-perturbation, a ghost manifold occurs where the dynamics approximate the original attractor, but drift occurs slowly along this manifold. As for ring attractors, they retain a similar shape on post-perturbations with any drift. This capability to retain its bump in the same location is the reason why ring attractors was a more suitable candidate for us. For the second part of the question, we refer to Gallistel et al. (2000) as reference to our model. The accumulator model here considers memory magnitude (the number representation) to be estimated from memory noise. This working as two separate systems is supported from an extension of the paper by Leslie et al. (2008) (calibration learning for small numbers vs realisation learning for large numbers). This is why we considered it as two separate systems (where we focus on subitization) instead of one system.
> > >
> > > Q2. How do the authors choose the various hyperparameters mentioned in Table 1? Have the authors tested the robustness of the results to changes in hyperparameters apart from the time constant?
> > >
> > > A: The hyper parameters were initially based on C. R. Laing et al. (2001). The parameters tested to ensure this works are capacitance, membrane time constant, input current and the connection weights. For each part of the sub-system (as per fig 1), these parameters are experimented and chosen based on how the sub-system responds (the firing rate pattern for different rates of poisson generator). This has been added to the manuscript to clarify the issue in Table 1.
> > >
> > > ### References
> > >
> > > Stanislas Dehaene and Laurent Cohen. Dissociable mechanisms of subitizing and counting: neu-
> > > ropsychological evidence from simultanagnosic patients. Journal of Experimental Psychology:
> > > Human perception and performance, 20(5):958, 1994.
> > >
> > > Mandler, G., & Shebo, B. J. (1982). Subitizing: An analysis of its component processes. Journal of Experimental Psychology: General, 111, 1‑22. Journal of Experimental Psychology: General, 111(1), 1-22. Retrieved from https://escholarship.org/uc/item/9fn27772
> > >
> > > Leslie, Alan M. ; Gelman, Rochel & Gallistel, C. R. (2008). The generative basis of natural number concepts. Trends in Cognitive Sciences 12 (6):213-218.
> > >
> > > Kutter, E.F., Dehnen, G., Borger, V. et al. Distinct neuronal representation of small and large numbers in the human medial temporal lobe. Nat Hum Behav 7, 1998–2007 (2023). https://doi.org/10.1038/s41562-023-01709-3
> > >
> > > Laing, C. R., & Chow, C. C. (2001). Stationary bumps in networks of spiking neurons. Neural Computation, 13(7), 1473–1494.
> > >
> > > C.R. Gallistel and Rochel Gelman. Non-verbal numerical cognition: from reals to integers. Trends
> > > in Cognitive Sciences, 4(2):59–65, 2000.
> > >
> > > Ságodi Á, Martín-Sánchez G, Sokół P, Park IM. Back to the Continuous Attractor. ArXiv [Preprint]. 2024 Nov 5:arXiv:2408.00109v2. PMID: 39130204; PMCID: PMC11312635.

---

> > > > ### Comment · Reviewer_EBGP · 2024-12-02
> > > >
> > > > I thank the authors for their response. I have carefully read the authors' comments and unfortunately, I think some more work is required to flesh this out into a complete submission. I encourage the authors to think carefully about the questions raised by reviewers here, and work towards improving the experimental results and justifying the model better. Hence, I will maintain my original assessment and score.

---

### Official Review · Reviewer_9ToH · 2024-11-09

**Soundness:** 2
**Presentation:** 2
**Contribution:** 2
**Rating:** 3
**Confidence:** 3

**Summary:**

The paper presents the AlloNet model, which applies allostatic control principles to spiking neural networks to maintain adaptable, persistent internal states. By extending the biological concept of allostasis, the authors enable a ring attractor network to dynamically align internal representations with external stimuli. The model is specifically applied to a subitization task, where it demonstrates the ability to align internal numerical representations with external stimuli through an attractor network’s localized bump of activity.

**Strengths:**

The authors provide a novel approach to applying allostatic principles to control internal representations within spiking neural networks. This approach may inspire further cross-disciplinary exploration of allostasis and neural network control. In particular, the use of allostasis as a tool of synchronizing the alignment between internal representations and external stimuli could make the model useful in applications requiring real-time adaptability, such as robotics or artificial agents interacting with unpredictable environments.

**Weaknesses:**

* The paper presents allostasis as a beneficial mechanism but does not sufficiently compare it to traditional homeostatic mechanisms or other neural adaptation frameworks. This limits understanding of when allostatic control is most useful or essential, making it difficult to gauge the model's full significance

* The experimental setup is limited to idealized, controlled tasks. Real-world applications, however, typically involve noisy, unpredictable inputs that can disrupt internal representations, which this model might struggle with. Current experiments do not address such robustness.

* The motivation for using a ring attractor with a "bump of activity" as the representation for the task of numerical cognition (e.g., subitizing) is not clear. What is the (systems) neuroscience evidence for this?

* The authors note that error rates increase with numerosity and time, but more in-depth insights into the reasons for these errors, such as bump instability, could clarify model limitations

**Questions:**

* Is the bump instability, in Fig. 5, influenced primarily by synaptic time constants, or are other factors like network noise equally contributory?

* It would be interesting to see the AlloNet benchmarked against other attractor models that handle drift, like ring attractor networks with stabilizing feedback (e.g., [Accurate Path Integration in Continuous Attractor Network Models of Grid Cells
](https://journals.plos.org/ploscompbiol/article?id=10.1371/journal.pcbi.1000291))

---

> ### Author Response · Authors · 2024-12-02
>
> ### Weaknesses
>
> W1. The paper presents allostasis as a beneficial mechanism but does not sufficiently compare it to traditional homeostatic mechanisms or other neural adaptation frameworks. This limits understanding of when allostatic control is most useful or essential, making it difficult to gauge the model's full significance.
>
> A: Thank you for pointing out on this. The allostatic aspect of the model is restricted to the fact that the numerosity acts as a variable set point to what would otherwise be a purely homeostatic mechanism. Further allostatic interaction (like prediction and other environmental effects) are not taken into account in the current model.
>
> W2. The experimental setup is limited to idealized, controlled tasks. Real-world applications, however, typically involve noisy, unpredictable inputs that can disrupt internal representations, which this model might struggle with. Current experiments do not address such robustness.
>
> A: Thank you for your comment on this. Some of the work on ring attractor’s capability to handle noise has been done in other works. These ring attractors (continuous attractors) are capable of encoding representations even in the presence of noise (behavioral accuracies) as per Klaus Wimmer (2014). These can be shown even for small networks where they may be considered as a continuous system that is robust to noise (Marcella Noorman et al (2024)). Regarding the perceptual noise, we agree with you that it required an input layer that could handle more complex visual inputs, however, we considered this as a next step in our work ,once the basic mechanism were understood.
>
> W3. The motivation for using a ring attractor with a "bump of activity" as the representation for the task of numerical cognition (e.g., subitizing) is not clear. What is the (systems) neuroscience evidence for this?
>
> A: The system for the number line we consider is linear for smaller numbers as portrayed in Leslie et al. (2008). This considers it as an accumulator system and thus we consider how we store these representations as a form of attractor in a Hopfield network. The structure we considered here is on the ring attractor since the arrangements of connections in this network allow us to give a form of magnitude (due to the bump activity required to move through sequential order).
>
> W4. The authors note that error rates increase with numerosity and time, but more in-depth insights into the reasons for these errors, such as bump instability, could clarify model limitations.
>
> A: The system uses a homeostasis system with the LGM and HGM perturbs the bump activity to move and this causes instability in the system to move the bump. The limitation of the model here is that when regaining stability (when there are no influence from the homeostasis system), the model may stabilise in a region close to our perception thus having some errors.
>
> ### Questions
>
> Q1. Is the bump instability, in Fig. 5, influenced primarily by synaptic time constants, or are other factors like network noise equally contributory?
>
> A: The bump instability is caused by perturbations of spike made by the LGM and HGM. This acts as inhibition on the ring attractor network. The spike pattern for these two regions (LGM and HGM) are influenced by the homeostasis network and this pattern is configured to ensure that these perturbations causes instability to kill the bump but does not kill the activity off completely.
>
> Q2. It would be interesting to see the AlloNet benchmarked against other attractor models that handle drift, like ring attractor networks with stabilizing feedback (e.g., Accurate Path Integration in Continuous Attractor Network Models of Grid Cells )
>
> A: Thank you for your interest in this aspect of the model. Currently, we have not benchmarked this against other models with respect to how it handles drift. These will be done for future work.
>
> ### References
>
> Leslie, Alan M. ; Gelman, Rochel & Gallistel, C. R. (2008). The generative basis of natural number concepts. Trends in Cognitive Sciences 12 (6):213-218.
>
> Wimmer K, Nykamp DQ, Constantinidis C, Compte A. Bump attractor dynamics in prefrontal cortex explains behavioral precision in spatial working memory. Nat Neurosci. 2014 Mar;17(3):431-9. doi: 10.1038/nn.3645. Epub 2014 Feb 2. PMID: 24487232.
>
> Noorman, M., Hulse, B.K., Jayaraman, V. et al. Maintaining and updating accurate internal representations of continuous variables with a handful of neurons. Nat Neurosci 27, 2207–2217 (2024). https://doi.org/10.1038/s41593-024-01766-5

---

> > ### Comment · Reviewer_9ToH · 2024-12-03
> > **Official Response by Reviewer**
> >
> > I appreciate the authors’ effort into addressing the raised concerns. The proposed approach is conceptually interesting in applying allostatic principles to neural network design, which I find to be a compelling direction for future exploration. However, I feel the paper, in its current form, does not sufficiently address these foundational and practical concerns to make a fully convincing case for its contributions. I encourage the authors to continue refining this work. I will keep my score.

---

### Official Review · Reviewer_Kxxg · 2024-11-10

**Soundness:** 2
**Presentation:** 2
**Contribution:** 2
**Rating:** 5
**Confidence:** 3

**Summary:**

The paper introduces AlloNet, a spiking neural network architecture that incorporates an allostatic control mechanism for the regulation of persistent states. The authors use a ring attractor network coupled with a Hammel model to achieve dynamic control of spatial changes in neuronal activity. The model is applied to a numerical cognition task (subitization) to demonstrate its ability to modulate the location of a bump of activity as a function of a reference input.

The main idea appears to be original and effective, however the relevance of the model, such as biological plausibility and comparison with other models could be better discussed.

**Strengths:**

1. Novelty of the Approach
Integrating an allostatic control mechanism into a spiking neural network architecture is a novel approach with potential implications for understanding self-regulation in neural systems.

2. Dynamic Control of Persistent States
The model successfully demonstrates the dynamic control of persistent states in response to environmental changes, a crucial aspect of cognitive processing.

3. Qualitative Reproduction of Behavioral Aspects
AlloNet qualitatively reproduces certain behavioral aspects of subitization, such as the relationship between reaction time and numerosity.

**Weaknesses:**

Limited Biological Plausibility: While the model draws inspiration from biological systems like the Hammel model for temperature regulation, the direct application of such a model to numerical cognition might oversimplify the underlying biological mechanisms.

Specificity of the Model: The paper focuses heavily on subitization as an application. It would be beneficial to explore additional cognitive tasks to demonstrate the generalizability of AlloNet.

**Questions:**

-I suggest to quickly introduce the Hammel model, for helping the reader.

-Could the authors elaborate on the biological plausibility of using the Hammel model, specifically in the context of numerical cognition? Are there any alternative biological mechanisms that might be more relevant?

-How does AlloNet compare to other models of numerical cognition in terms of performance and biological plausibility?

-What are the potential implications of this model for understanding cognitive deficits or disorders related to numerical processing?

Minor

-Caption of Fig5 should be more clear. E.g. what is panel B exactly, make clearer what colors refer to. (are bumps of different ring attractors?).

-References to figure 4 should be “Fig4” rather than just “4”.

-There is a missing dot at the end of the abstract, and at the end of figure 4 caption.

-fig1d is not readable

---

> ### Author Response · Authors · 2024-12-02
>
> ### Weaknesses
> W1. Limited Biological Plausibility: While the model draws inspiration from biological systems like the Hammel model for temperature regulation, the direct application of such a model to numerical cognition might oversimplify the underlying biological mechanisms.
>
> A: Thank you for pointing this out. In our case, we look at Leslie et al. (2008) where our focus lies on the early stages of the number line, specifically the range that can be considered linear (1–4). Gunderson et al. (2012) further establish that the linear number line is closely related to spatial skills, which is why our approach emphasizes subitization. The current model is purposefully designed within this context to showcase its capability in demonstrating visual-spatial skills. This provides a foundational step for extending the model toward robotic applications, particularly in enabling hand-eye coordination for perception-based manipulation tasks.
>
> W2. Specificity of the Model: The paper focuses heavily on subitization as an application. It would be beneficial to explore additional cognitive tasks to demonstrate the generalizability of AlloNet.
>
> A: Thank you for pointing out the limited application provided by the current model in our case. Due to time limitations, we are unable to add the experiments planned for future work. In this case, we intend to apply the model to a robotic application. 1. To feed the network with processed sensory data as a form of perception. 2. Connect the ring attractor where the robotic manipulation acts in response to the bump activity. As per weakness 1, I have stated the motivation behind how robotic use cases will be related to this paper’s experiments.
>
> ### Questions
>
> Q1. I suggest to quickly introduce the Hammel model, for helping the reader.
>
> A: I have added a quick introduction to this on (Line 57-58) to give a brief initial explanation on the Hammel Model.
>
> Q2. Could the authors elaborate on the biological plausibility of using the Hammel model, specifically in the context of numerical cognition? Are there any alternative biological mechanisms that might be more relevant?
>
> A: In this work, we based it off on Leslie et al. (2008) and Gallistel (2000). Here, we consider the accumulator model where the number counted is bidirectionally mapped to the magnitude of the accumulation. The accumulation here can be considered as a feedback system which goes in line with what the Hammel Model provides. The Hammel Models allow us to view homeostasis as a working mechanism of a negative feedback mechanism.
>
> Q3. How does AlloNet compare to other models of numerical cognition in terms of performance and biological plausibility?
>
> A: We have not tested our model in this aspect other than in comparison with human performance. This will be further explored and done in future work.
>
> Q4. What are the potential implications of this model for understanding cognitive deficits or disorders related to numerical processing?
>
> A: Thank you for this pertinent question. Indeed, we are interested about the potential implication of this model in disorders like dyscalculia. For example Mazzocco et al. (2011) shows Approximate Number System (ANDS) impairment in people with dyscalculia (poorer precision). The relationship persisted even when controlling for domain-general abilities. Our model associates the capacity of subitizing (ANS) with the stability of the bump once it reaches the corresponding numerosity. This itself depends on the synaptic constants, among other parameters. Future work would be focused on investigating if such dynamic mechanisms produce useful predictions that could be compared with human subject data.
>
> Minor Detail Clarification:
>
> Q1. Caption of Fig5 should be more clear. E.g. what is panel B exactly, make clearer what colors refer to. (are bumps of different ring attractors?).
>
> A: The colours are encoded towards each number (as shown on the left). It defines the centroid of the bump activity (i.e. where the mean of the activity is). I have added in additional details to explain this in Fig 5 caption.
>
> Q2. References to figure 4 should be “Fig4” rather than just “4”.
> There is a missing dot at the end of the abstract, and at the end of figure 4 caption.
>
> A: I have added minor clarifications to Figure 4 and the dot to caption of Fig 5.
>
> Q3. fig1d is not readable
>
> A: Thank you for pointing this out. To address better clarity, I have increased the text size in the figure. Further changes made are on figure 1A being bigger to improve clarity.

---

> ### Author Response · Authors · 2024-12-02
>
> ### References
>
> Gunderson, Elizabeth & Ramirez, Gerardo & Levine, Susan & Beilock, Sian. (2011). The Role of Parents and Teachers in the Development of Gender-Related Math Attitudes. Sex Roles. 66. 153-166. 10.1007/s11199-011-9996-2.
>
> Leslie, Alan M. ; Gelman, Rochel & Gallistel, C. R. (2008). The generative basis of natural number concepts. Trends in Cognitive Sciences 12 (6):213-218.
>
> Mazzocco, M. M., Feigenson, L., & Halberda, J. (2011). Impaired acuity of the approximate number system underlies mathematical learning disability (dyscalculia). Child development, 82(4), 1224-1237.
>
> C.R. Gallistel and Rochel Gelman. Non-verbal numerical cognition: from reals to integers. Trends in Cognitive Sciences, 4(2):59–65, 2000.

---

### Meta-Review · Area_Chair_cp3g · 2024-12-22

**Metareview:**

This work introduces a spiking neural network that that dynamically adjusts representation based on environmental inputs, drawing inspiration from the body’s temperature regulation system and incorporating the concept of allostasis.  The reviewers praised the paper for its originality and innovative introduction of physiological concepts into neural network modeling. However, they raised concerns about biological plausibility, thoroughness of the experimental demonstrations, and lack of clarity in the manuscript, which ultimately left the paper below the threshold for acceptance. I regret that it cannot be accepted to this year's meeting, but I with the authors the best of luck in submitting it for publication elsewhere.

**Additional Comments On Reviewer Discussion:**

The authors added clarifying information during the rebuttal period, but the reviewers did not feel that they it sufficiently addressed the foundational concerns raised during the reviews.

---

### Decision · Program_Chairs · 2025-01-22

Reject